# Identification of Putative Novel Rotavirus H VP7, VP4, VP6 and NSP4 Genotypes in Pigs

**DOI:** 10.3390/v16010068

**Published:** 2023-12-30

**Authors:** Elena Ferrari, Greta Vignola, Cristina Bertasio, Chiara Chiapponi, Giovanni Loris Alborali, Vito Martella, Maria Beatrice Boniotti

**Affiliations:** 1Istituto Zooprofilattico Sperimentale della Lombardia e dell’Emilia Romagna “Bruno Ubertini”, Via Bianchi 7/9, 25124 Brescia, Italy; greta.vignola@izsler.it (G.V.); cristina.bertasio@izsler.it (C.B.); chiara.chiapponi@izsler.it (C.C.); giovanni.alborali@izsler.it (G.L.A.); mariabeatrice.boniotti@izsler.it (M.B.B.); 2Department of Veterinary Medicine, University of Bari Aldo Moro, 70010 Valenzano, Italy; vito.martella@uniba.it

**Keywords:** Rotavirus H, pig, enteritis, mixed infection, genotyping, genetic variability

## Abstract

Rotavirus H (RVH) has been detected in humans, pigs and bats. Recently, RVH infections were reported in different porcine farms worldwide, suggesting epidemiological relevance. However, to date, the genome information of RVH strains has been limited due to the scarcity of deposited sequences. This study aimed to characterize the VP7, VP4, VP6 and NSP4 genes of RVHs from 27 symptomatic pigs, in Italy, between 2017 and 2021. RVH genes were amplified via RT-PCR using specific primers, and the amplicons were sequenced. By coupling the data generated in this study with the sequences available in the databases, we elaborated a classification scheme useful to genotype the VP7, VP4, VP6 and NSP4 genes. The nucleotide identity and phylogenetic analyses unveiled an impressive genetic heterogeneity and allowed the classification of the Italian RVH strains into 12G (VP7), 6P (VP4), 8I (VP6) and 8E (NSP4) genotypes, of which 6I, 5E and the totality of the G and P genotypes were of novel identification. Our data highlight the high genetic variability of the RVH strains circulating in pigs and underline the importance of a robust classification system to track the epidemiology of RVHs.

## 1. Introduction

Rotaviruses (RVs) represent one of the major etiological agents of enteritis in humans and animals worldwide. Based on the genetic diversity of the intermediate capsid protein (VP6), RVs are currently classified into nine groups (A–L) (https://talk.ictvonline.org/taxonomy/, accessed on 1 September 2023). The genome comprises 11 segments of double-stranded RNA encoding for six structural (VP1–VP4, VP6 and VP7) and five or six non-structural proteins (NSP1–NSP5/NSP6) [1]. The outer proteins VP4 and VP7 mediate the interaction with host cell receptors and induce immunity protection by eliciting neutralizing antibodies [2]. VP6 is involved in the transcription activity of the double-layered virion [2]. The NSP4 protein is an enterotoxin that causes intracellular calcium imbalance and induces secretory diarrhea [2]. Due to their role in the host range restriction and pathogenicity, the gene segments encoding these proteins are often molecularly characterized [3,4].

RV infections are highly prevalent in pig herds, frequently associated with acute diarrhea in young pigs, representing an important concern for the swine industry. The RV groups most commonly associated with enteric disease in pigs are RVA, RVB, RVC and RVH [5,6,7].

RVH was first identified in China in sporadic cases in human patients with gastro-enteritis in 1987 and 1988 [8]. In 1994 in Beijing and 1997 in Shijiazhuang, RVH caused large gastroenteritis outbreaks [9]. RVH was initially named as new adult diarrhea RV (ADRV-N) [10,11]. In pigs, RVH was first described in fecal samples of animals affected by diarrhea between 1991 and 1995 in Japan [12]. Between 2008 and 2022, the presence of porcine RVH was also found in the United States [13], Brazil [14], South Africa [15], Vietnam [16] and China [17] as well as some European countries, such as Spain [18], Italy [7] and Russia [19]. In the period 2016–2022, the rates of detection of RVH in symptomatic pigs ranged between 9 and 14% in Spain [18], Italy [7], Brazil [20] and China [17]. These values indicate that RVH is relatively widespread in swine populations, although the impact of RVH on porcine herds in terms of costs and animal health has not been assessed.

Previous studies reported that RVH infects mostly adult pigs in combination with other RV groups [7,20,21]. In addition to co-infections, infections by multiple strains of the same group (e.g., RVA) were also described [22,23]. While co-infections have been reported to increase the severity of diarrhea [24,25], the clinical impact of co-infections is not known.

Genome sequencing is crucial to understand the evolutionary and epidemiological relationship among different RV strains and the adoption of a robust classification scheme is essential to facilitate data sharing and to understand promptly the origin of genome segments, thereby unveiling events of interspecies transmission, among animals and from animals to humans, eventually coupled with reassortment of RV genome segments [26]. To date, a uniform classification system for all 11 gene segments has been established by the Rotavirus Classification Working Group (RCWG) only for RVA strains but not for the other RV groups [27]. Despite the relatively low number of complete genomic sequences of RVH available, recently, a system for assigning genotypes, similar to the classification scheme adopted for RVA, has been suggested [21]. The RVH classification scheme has proposed 10G, 6P, 6I, 3R, 4C, 7M, 6A, 2N, 4T, 6E and 3H-genotypes for the VP7, VP4, VP6, VP1, VP2, VP3, NSP1, NSP2, NSP3, NSP4 and NSP5 genes, respectively. This genotype system is based on the alignment of complete ORF sequences for each gene of RVHs available in GenBank and the adoption of cut-off values and additional criteria for partial ORF sequences [27]. Based on the analysis of RVH sequences, it is clear that human strains are genetically highly homogeneous, whilst porcine RVH strains are genetically diverse, even in restricted geographical areas [21], suggesting that human RVH originated from a recent bottleneck event from an unidentified animal host. However, the limit of this RVH classification scheme is that it relies on a small database. For these reasons, our study aimed to collect additional RVH sequences in order to implement the classification system. These data serve as a foundation for developing accurate molecular diagnostics and conducting comprehensive epidemiological investigations.

To achieve this purpose, we determined the nucleotide sequences of genes VP7, VP4, VP6 and NSP4 from 27 RVH-positive stool specimens collected from Italian porcine herds in the period 2017–2021. The RVH classification scheme was optimized using the larger sequence data set re-calculating the nucleotide cut-off values previously proposed [21] for the classification of VP7, VP4, VP6 and NSP4 genotypes [27]. Through this process, we identified a great multiplicity of novel RVH genotypes circulating in pigs.

## 2. Materials and Methods

### 2.1. Samples

Between January 2017 and December 2021, 27 fecal samples from pigs with enteric disease, obtained as part of routine analyses conducted by the Istituto Zooprofilattico Sperimentale della Lombardia ed Emilia Romagna, were determined as RVH-positive using a previously established RT-qPCR protocol [7]. These specimens were collected from 23 different porcine farms across Italy: 17 from Lombardia, 2 from Veneto, 2 from Emilia Romagna, 1 from Piemonte and 1 from Umbria. Three farms were sampled twice in the same year (farm 2, 12, 13), while one farm was sampled twice in two different years (farm 4). All the farms were industrial, with 19 managed for farrow to weaning and 8 for weaning (Table 1). All the pigs were fed with commercialized food and none were subjected to an anti-RVA vaccination plan. Age data and the status of infection by RV groups, tested by RT-qPCR [7], are reported in Table 1.

### 2.2. RNA Extraction and Amplicon Generation

Double-stranded RNA was extracted from 200 µL of 10% fecal suspension in a minimum essential medium using QIAzol Lysis Reagent (QIAGEN, Hilden, Germany) according to the manufacturer’s instructions. The extracted RNA underwent a denaturation step at 95 °C for 5′, and RT-PCR was performed using the SuperScriptIV One step kit (Invitrogen, Waltham, MA, USA) according to the manufacturer’s protocol using custom primers. The primers employed (Table 2) were designed based on the full-length sequences of porcine RVH VP7, VP4, VP6, and NSP4 available in the NCBI GeneBank database (Appendix A). The PCR products were purified via the Nucleospin^®^ gel kit (Macherey-Nagel, Düren, Germany). The concentration and the quality of the obtained DNA were assessed using the Nanoquant Infinite M200 spectrophotometer (Tecan, Männedorf, Switzerland).

### 2.3. Sequencing

Purified DNA products from each sample were pooled and quantified using a QuantiFluor^®^ ONE dsDNA kit and a Quantus Fluorometer (Promega, Fitchburg, MA, USA). The libraries were generated with an Illumina DNA Prep (M) tagmentation kit and sequenced on an Illumina MiniSeq platform with 2 × 150-bp paired end reads (Illumina, San Diego, CA, USA). Adapters were automatically removed from FASTQ files. Raw reads were filtered to remove low-quality bases (Phred score < 30) and trimmed to remove residual sequencing adapters using Trimmomatic (v 0.39).

The reads were de novo assembled into contigs by CLC Genomic Workbench (v.23.0.5, QIAGEN, Hilden, Germany), and SPAdes (v.3.15.5) [28] assemblers. Contig sequences that were identified by both assemblers and that showed a minimum average coverage of 30 were employed for the phylogenetical analyses. The gene sequences which resulted as incomplete through the Illumina approach were re-sequenced using a BigDye Terminator v1.1 Cycle Sequencing kit on an automated ABI Prism 3500 × l Genetic Analyzer (Thermo Fisher Scientific, Waltham, MA, USA) using the primers reported in Table 2. The identity of each segment was confirmed via BLAST analysis using the National Center for Biotechnology Information GenBank Tool (NCBI, https://www.ncbi.nlm.nih.gov/, accessed on 3 July 2023). The sequences generated in our study were deposited into GenBank under accession numbers from OR817801 to OR817920.

### 2.4. Genotyping and Phylogenetic Analyses

The nucleotide sequences were aligned with known genotypes of porcine and human RVH strains available in the GenBank database (Appendix A) using the ClustalW software (v.2.1) implemented in BioEdit, version 7.2.5 [29]. Genetic distances were calculated using Kimura’s two-parameter correction at the nucleotide level, performed with MEGA v.11 [30]. The nucleotide cut-off values for genotype assignment were established based on a nearly full-length ORF for each gene (ORF > 94%) of reference and Italian sequences (Appendix A). Pairwise identity frequency graphs, obtained via genetic distances, were constructed by plotting all the calculated pairwise identities on a graph with the percentage of identity on the *x*-axis and the frequency of each of the calculated pairwise identities on the *y*-axis. The most appropriate cut-off was defined as the value that separates the intra-genotype identities and the inter-genotype identities in order to avoid overlaps between different genotypes. In the case of sequences with partial ORF (<94%), genotype assignment was performed using values 2% higher than those calculated for complete ORFs according to the RCWG criteria for RVA [27]. However, only sequences encompassing at least 80% of the ORF length were genotyped. Minimum and maximum values of nucleotide identity between genotypes, intra-genotype (Appendix A), and between the reference and the Italian strains were calculated (Appendix A). The phylogenetic trees were constructed on the partial ORF of VP7 (81%), VP4 (94%), VP6 (92%), and NSP4 (88%) genes using the maximum-likelihood method with 500 bootstrap replicates. The best-fit nucleotide model for each gene dataset was selected based on the lowest Bayesian Information Criterion (BIC) score upon the model testing in Mega v.11 software [30]. The selected best-fit models were the Tamura 3 parameter (T92) model with the discrete Gamma distribution (G) and Invariant sites (I) for VP6 and VP7, Hasegawa-Kishino-Yano (HKY) with G+I for NSP4, and General Time Reversible (GTR) model with G+I for VP4 sequences.

### 2.5. Recombination Analyses

Recombination events were evaluated using the Recombination Detection Program (RDP) (v.4.101) with RDP, GENECOV, Bootscan, MaxChi, Chimera, SiSscan, and 3Seq algorithms [31]. The recombination analysis was conducted using strains of this study and human and porcine RVH (Appendix A), RVA, RVB and RVC sequences available in Genbank. Similarity plot analysis was conducted using the SimPlot software (v. 3.5.1) with a sliding window of 200 bp (step: 20 bp).

## 3. Results

### 3.1. VP7

Eight samples (ITA/48625-1/2018, ITA/101803/2018, ITA/146160/2018, ITA/345422/2018, ITA/25635-4/2020, ITA/56594/2020, ITA/31399/2020 and ITA/42091-1/2020) presented multiple alleles of the VP7 gene, suggesting that different strains were circulating and infecting pigs within the same farm. The VP7 nucleotide cut-off, calculated by including 28 Italian and 31 reference sequences with nearly full-length ORF, was established at 86%, consistent with the previous proposal for this gene [21]. Based on these cut-off values, all the sequenced strains showed a percentage of identity below the proposed threshold, suggesting that they could belong to novel genotypes. As a results, we identified twelve potential new genotypes (G11-G22) (Figure 1). Fourteen strains, classified as G13 and G18 genotypes, were more related to Japanese strains belonging to G5 and G10 genotypes. Another seven strains, which were grouped into four different genotypes (G11-G14-G17-G19), shared a common ancestor with G3 from Brazil and Spain. Fourteen sequences were phylogenetically distant from reference strains and, based on our proposed nucleotide cut-off, were classified into five different genotypes (G15-G16-G20-G21-G22). Identical G-genotypes were detected in samples collected from the same farm during the same year (farms 2, 12, 13). Conversely, distinct G-genotypes were identified in samples ITA/101803/2018 (G13-G22) and ITA/105389/2019 (G20) collected from farm 4 in different years (2018 and 2019, respectively) (Table 3). Sample strain ITA/105389/2019 shared a 70% and 83% nucleotide identity with sample strain ITA/101803/2018.

### 3.2. VP4

Three out of twenty-seven fecal samples (ITA/48625-1/2017, ITA/76963/2018 and ITA/165825/2021) exhibited mixed VP4 sequences. The genotyping threshold for the VP4 gene was adjusted from the previously proposed 86% [21] to 87%. Upon comparing 94% of the ORF sequences, it was observed that all 26 Italian strains shared a percentage of identity lower than 87% with the reference strains, suggesting that they could belong to novel genotypes (P7-P12). Similarly, the phylogenetic tree showed that Italian strains are not closely related to reference VP4 sequences (Figure 1). Strain ITA/51105-6/2020 shared a nucleotide identity above the 87% cut-off with only 1 sequence of P10 genotype (ITA/14193-5/2020), but phylogenetically, it belonged to a different clade. For these reasons, the genotype assignment remained unclear. Interestingly, the alignment of the region of 1425 nucleotides in length (from position 945 to 2370) of ITA/51105-6/2020 and ITA/14193-5/2020 showed a high degree of similarity (97%). Analysis conducted via the RDP software (v.4.101) with sequences of porcine RVH strains evidenced the presence of significant recombination events within ITA/51105-6/2020 and ITA/14193-5/2020 (Table 4).

Despite both the recombinants shared the larger fraction of the sequence, the origin of the minor fraction of VP4 gene of ITA/51105-6/2020 remained unknown, while that of ITA/14193-5/2020 was probably derived from four P10 strains (ITA/154488/2018, ITA/48625-1(I)/2017, ITA/101803/2018 and ITA/305471/2017). The similarity plot analysis confirmed the RDP results, evidencing the position of the breakpoint at 945 bp of the gene sequence (Appendix A). Samples collected within the same year and from the same farm (2, 12, 13) were infected by strains of the same genotype (Table 3). On the contrary, samples collected in the same location but during two different years showed the presence of different genotypes.

### 3.3. VP6

Five fecal samples (ITA/76963/2018, ITA/25635-4/2020, ITA/56594/2020, ITA/42091-1/2020, ITA/243418/2021) presented mixed VP6 sequences, suggesting the possibility of a co-infection involving multiple RVH strains. The cut-off value for differentiating among VP6 genotypes was calculated based on the frequency distribution of pairwise sequence identities among 25 Italian porcine strains and 59 reference strains from both human and porcine origin. Through this analysis, the nucleotide threshold was set to 88%, an increase of 1% from the previously proposed value of 87% [21]. By applying this updated value, we identified eight different genotypes (I7-I12, I3 clade 1 and clade 2), of which only two (I3 clade 1 and clade 2) were found to be phylogenetically related to strains of the I3 genotype from Spain (VC-19, VC-29 and VC-36) [18] (Figure 1).

However, the Italian strains that cluster with Spanish SP-VC29, SP-VC36 and SP-VC19 constitute two separate clades distinct from those of other I3 strains (from Brazil and South Africa). These data suggest that these strains could belong to novel genotypes.

Samples collected from the same farm (farm 2, 12, 13) during the same year (2020), as well as those collected in different years (farm 4, 2018 and 2019), were found to be infected by the same genotype I (Table 3).

### 3.4. NSP4

Only one sample presented more alleles of the NSP4 gene. Nearly complete ORFs (96%) of 22 NSP4 Italian and 30 reference strains were aligned, and the nucleotide cut-off was calculated. For this gene, the previously defined nucleotide cut-off (83%) was increased by 2% (to 85%). Six strains (ITA/220515/2017, ITA/43471/2020, ITA/154488/2018, ITA/243418/2021, ITA/355703/2019 and ITA/138298/2019) exhibited a nucleotide identity below the cut-off value for all known genotypes, suggesting their classification into five novel genotypes (E7-E11). Five strains (ITA/42091-1/2020, ITA/48625-1(II)/2017, ITA/116154/2021, ITA/101803/2018, ITA/305471/2017) shared a nucleotide identity above the cut-off value with Brazilian and South African E3 strains, but below the cut-off with another E3 strain, VC-29, from Spain. Furthermore, the phylogenetic tree positioned these strains in two distinct clades (E3 clade 1-E3 clade 2) (Figure 1). E3 clade 1 was distinct from all E3 reference sequences, while E3 clade 2 was related to the Spanish E3 strain VC-29. Similarly, 13 sequences (ITA/14193-5/2020, ITA/50439/2020, ITA/51105-6/2020, ITA/66533/2020, ITA/56594/2020, ITA/140201/2019, ITA/31399/2020, ITA/48625-1(I)/2017, ITA/294885/2019, ITA/146160/2018, ITA/48625-1(III)/2017, ITA/103398/2019 and ITA/105389/2019) that were related to the Spanish strain SP-VC36, previously described as E4 genotype [18], belonged to a different clade (E4 clade 1) of other E4 genotypes (USA/MN9.65 and Spanish VC18-VC19).

Strain ITA/84987/2021 shared percentages of identity above the cut-off with only a few E4 clade 1 sequences, so its classification is not well defined.

Distinct E-genotypes were detected in samples collected from the same farms during the same year or in different years (farms 2, 4, 13). Only in farm 12, which was sampled twice during the same year, the sequenced strains belonged to the same genotype (E4 clade 1) (Table 3).

## 4. Discussion

To date, a substantial amount of information regarding the evolution of RV group A is available due to the high number of genomic sequences deposited in Genebank and the presence of a consolidated genotyping system (RCWG) [27]. However, data on the genomic classification of other RV groups are poor and often inconsistent due to the absence of a standardized genotyping method. In particular, RVH, discovered fairly recently, in both diarrheic and healthy pigs, needs a detailed analysis in terms of genomic variability, host specificity, and geographic distribution. Therefore, this study sequenced and genotyped the genomic segments VP7, VP4, VP6 and NSP4, which are involved in virus infectivity and pathogenesis. These data were used for the validation and improvement of the classification system recently proposed [21]. The analyses were performed on the nearly full-length sequences of genes VP7, VP4, VP6 and NSP4, which were amplified by PCR with specific primers and sequenced by the next-generation method or, in some cases, by the Sanger method. In this study, attempts were made to characterize and genotype Italian porcine RVH strains, updating the VP7, VP4, VP6, and NSP4 cut-off values previously proposed [21]. For the gene VP7, the same cut-off value previously defined was established [21]. For the other genes, the values were slightly increased. In the case of partial ORFs (<94% ORF), we applied the criteria established by RCWG for RVA, which recommend a minimum ORF length (>50%) and threshold (above 2% of the appropriate cut-off for complete ORFs). For a more reliable classification, we included only sequences containing more than 80% of the ORF.

Based on these criteria, we identified 12G, 6P, 8I and 8E-different genotypes in porcine herds in Northern Italy during the period 2017–2021. Interestingly, many of the described genotypes were classified as potential new types and were not related to any reference strain. Only in the case of the VP6 and NSP4 genes were some strains genetically related to the reference strains of established genotypes.

Based on nucleotide identity and phylogenetic analyses, most of the VP6 sequences were closely related to the Spanish strains VC-19, VC-29 and VC-36, which were previously classified as I3 genotype [18]. However, they were phylogenetically distant from other I3 strains from Brazil and South Africa. These data suggest that the Spanish and Brazilian/South African strains may likely be classified into two distinct variants of the I3 genotype.

Similarly, NSP4 analyses showed that some sequences belonged to distinct clades of the E3 genotype, while other strains seemed to belong to different clades of E4 genotypes. However, with an increased number of sequences and updated nucleotide cut-off values for these genes, both genotype assignment and phylogenetic relationships are now better defined. Therefore, strains that were previously grouped in the same genotype cluster may be re-classified as belonging to a novel genotype/variant group. For this reason, sequencing and analyzing more RVH strains are necessary to provide robustness to the genotype classification system.

For NSP4 ITA/84987/2021 and VP4 ITA/51105-6/2020 sequences, genotype assignment was not achieved since nucleotide identity was above the threshold only with a few sequences of the same genotype.

In particular, among strains of porcine RVH, we identified two recombinants of VP4 (ITA/51105-6/2020 and ITA/14193-5/2020). These two strains were collected from the same farm in the same year and might have originated from P10 strains or an unknown genotype. However, our analyses did not evidence recombinations among strains of other RV groups, events less frequent but described recently in NSP3 between RVH and RVC strains [19,21]. However, aside from recombination, other events, such as reassortment and mutations, may have contributed to the high variability of strains. These events are common in rotaviruses due to their genome features being segmented and RNA-based [2].

To compare the variability of the strains from this study against reference strains collected worldwide (Japan, USA, South Africa, China, Spain), we calculated the range of nucleotide identity (Appendix A). Such analyses showed that the Italian strains, collected within a restricted geographical area over a limited timeframe (4 years), shared a sequence identity comparable to that observed among reference strains collected over a more extended period (Appendix A).

The wide variability observed among the Italian strains could be related to the high density of intensive swine farms, typical of the north of Italy. Moreover, the detection of multiple gene alleles within the same sample, as observed in some cases, would suggest the simultaneous circulation of different RVH strains within herds, which could favor reassortment and recombination events, further increasing the genetic variability of circulating strains. Finally, it is important to note that the NGS sequencing method employed herein was applied to PCR amplicons of gene segments using degenerated primers specific for the different RVH genes. With this approach, it cannot be excluded that specific gene alleles may be preferentially amplified in PCR. In addition, in some cases, a gene sequence was obtained by a combination of sequencing technique to confirm the data. It should be noted that when Sanger sequencing was applied, it was possible to determine only the dominant sequence whilst mixed infections resulted in poor-quality electropherograms.

This study highlighted the great variability among RVH strains, but it does not explore the correlation between specific genotypes and clinical symptoms, both because the collected samples exhibited co-infections with other RV groups (Table 1) and because the sample size was too small to achieve statistically significant correlation. Further investigations are needed to explore these aspects.

## 5. Conclusions

In this study, we genotyped VP7, VP4, VP6 and NSP4 gene segments from porcine RVH strains in Italy, proposing an update of the classification scheme for RVH with new nucleotide cut-offs for VP4, VP6 and NSP4 segments. Using this scheme, we provided a snapshot of the variability of RVH strains in porcine herds. Our analyses highlighted that multiple RVH genotypes circulated in swine herds over a limited timeframe. Therefore, increasing the number of RVH sequences and subsequently updating and standardizing the classification system could help in identifying the predominant genotypes, updating the diagnostic assay, and tracing the evolution of porcine RVH.

## Figures and Tables

**Figure 1 viruses-16-00068-f001:**
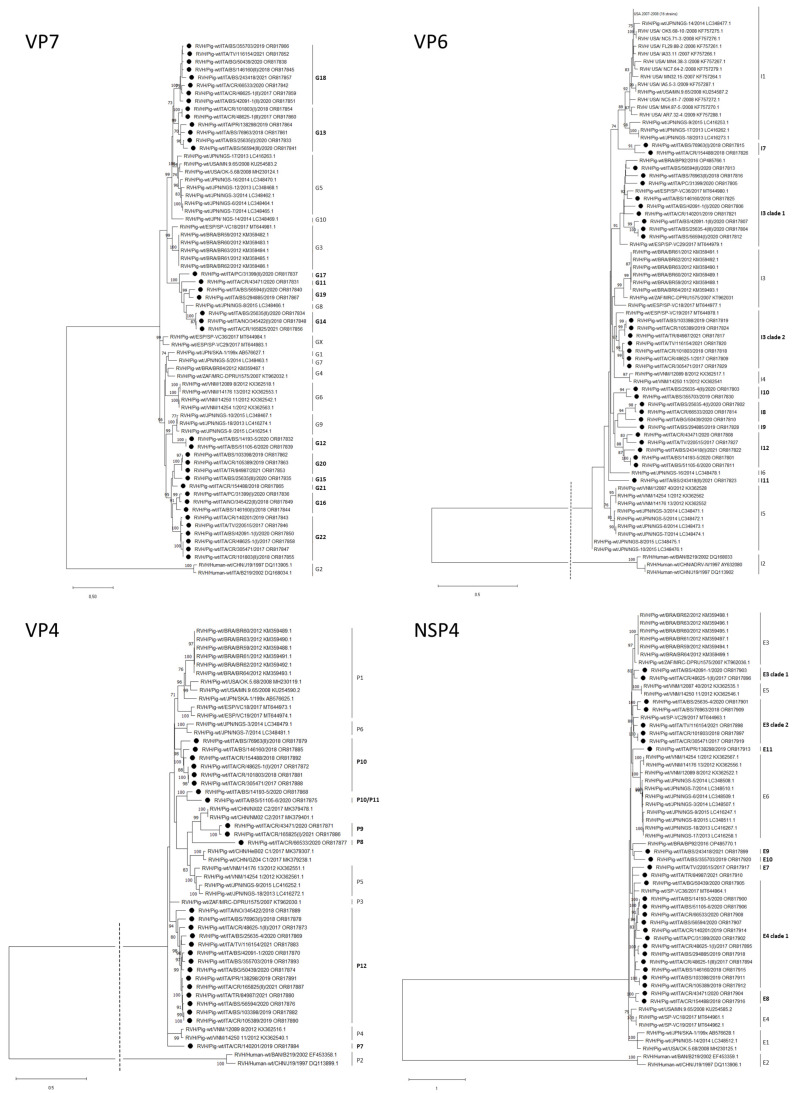
Phylogenetic trees constructed from a partial open reading frame of VP7, VP6, VP4 and NSP4 genes of RVH strains. Phylogenetic trees were constructed via the maximum likelihood method. Statistical support was provided via the bootstrapping of 500 pseudo-replicates. Bootstrap values above 70 are given at each branch node. Black circles represent the RVH strains analyzed in this study. Genotypes are specified on the right. The outgroup branch length for VP6 and VP4 was shortened (dotted line) due to its excessive length.

**Table 1 viruses-16-00068-t001:** Data on porcine farm management and RV infection status.

Strain	Management of Herd	Age	RVA	RVB	RVC	RVH
RVH/Pig-wt/ITA/CR/48625-1/2017	Farrow to weaning	weaning	+	+	+	+
RVH/Pig-wt/ITA/TV/220515/2017	Farrow to weaning	post-weaning	+	+	+	+
RVH/Pig-wt/ITA/CR/305471/2017	Farrow to weaning	fattening	+	+	+	+
RVH/Pig-wt/ITA/BS/76963/2018	Farrow to weaning	fattening	+	+	+	+
RVH/Pig-wt/ITA/CR/101803/2018	Farrow to weaning	weaning	+	+	+	+
RVH/Pig-wt/ITA/BS/146160/2018	Farrow to weaning	weaning	+	+	+	+
RVH/Pig-wt/ITA/CR/154488/2018	Farrow to weaning	fattening	+	+	+	+
RVH/Pig-wt/ITA/NO/345422/2018	Farrow to weaning	fattening	+	+	+	+
RVH/Pig-wt/ITA/BS/103398/2019	Farrow to weaning	weaning	+	+	+	+
RVH/Pig-wt/ITA/CR/105389/2019	Farrow to weaning	suckling	+	+	+	+
RVH/Pig-wt/ITA/PR/138298/2019	Weaning	post-weaning	+	+	+	+
RVH/Pig-wt/ITA/CR/140201/2019	Farrow to weaning	fattening	+	+	+	+
RVH/Pig-wt/ITA/BS/294885/2019	Weaning	fattening	+	+	+	+
RVH/Pig-wt/ITA/BS/355703/2019	Farrow to weaning	fattening	+	+		+
RVH/Pig-wt/ITA/BS/14193-5/2020	Farrow to weaning	post-weaning			+	+
RVH/Pig-wt/ITA/BS/25635-4/2020	Weaning	weaning	+	+	+	+
RVH/Pig-wt/ITA/PC/31399/2020	Farrow to weaning	post-weaning	+	+	+	+
RVH/Pig-wt/ITA/BS/42091-1/2020	Weaning	fattening	+	+	+	+
RVH/Pig-wt/ITA/CR/43471/2020	Weaning	fattening	+	+	+	+
RVH/Pig-wt/ITA/BG/50439/2020	Weaning	fattening	+	+	+	+
RVH/Pig-wt/ITA/BS/51105-6/2020	Farrow to weaning	weaning	+	+	+	+
RVH/Pig-wt/ITA/BS/56594/2020	Weaning	weaning	+	+	+	+
RVH/Pig-wt/ITA/CR/66533/2020	Farrow to weaning	fattening			+	+
RVH/Pig-wt/ITA/TR/84987/2021	Weaning	weaning	+	+	+	+
RVH/Pig-wt/ITA/TV/116154/2021	Farrow to weaning	fattening	+		+	+
RVH/Pig-wt/ITA/CR/165825/2021	Farrow to weaning	post-weaning			+	+
RVH/Pig-wt/ITA/BS/243418/2021	Farrow to weaning	weaning			+	+

“+ “indicates positivity to RV groups.

**Table 2 viruses-16-00068-t002:** Primers used for the amplification of the VP7, VP4, VP6 and NSP4 segments.

Gene Target	Primer	Sequence 5′-3′	Position
VP7	PoRVH_VP7 F1	GCC ATG TTG TTC CTA CTA AYC	12–32
	PoRVH_VP7 R2	GAT GTA AYG GAT TTC TCR ACG TT	781–803
VP4	PoRVH_VP4 F1	CAA GAR AAA TTR GAT CGY GAG	106–126
	PoRVH_VP4 R1	TGA CCA CAY TTY CTT GTT GG	1051–1070
	PoRVH_VP4 F2	TGG ATG ATT GAY TCA GGA TTT AA	955–977
	PoRVH_VP4 R3	GAC ATT ATG CCT GAA GTY AGA TC	1711–1733
	PoRVH_VP4 F4	GCA ATT GTT CCA GCT GAT GC	2089–2108
	PoRVH_VP4 R4	CTA CAG CAT ATT CTG CAA GAT G	2227–2248
	PoRVH_VP4 R2	ACT AAT GYC ACT ACR GTC TAT G	2498–2519
VP6	PoRVH_VP6_F1	TAC AAG TGA CCC ACA AGG ATG	14–34
	PoRVH_VP6_R1	ACA GGT ATR TTA TTT GGA GGC T	732–753
	PoRVH_VP6_F2	AGT ACC ATG TTC AGG AGT AAT G	658–679
	PoRVH_VP6_R3	GTT GCA ACA ATY CTT CCA CC	821–837
NSP4	PoRVH_NSP4 F1	ATC AAA GTM ACG ATG GAG CAC	1–21
	PoRVH_NSP4 R2	TTG CGC AAG GGT GRA CAC T	704–722

**Table 3 viruses-16-00068-t003:** Genotypes for each gene of the RVH strains obtained in the study.

Strains	Region	Farm	Genotype (N)														
			VP4	VP7	VP6	NSP4	VP4 (N = 6)		VP7 (N = 12)			VP6 (N = 8)			NSP4 (N = 8)		
RVH/Pig-wt/ITA/TV/220515/2017	Veneto	1	-	1	1	1	-		G22			I12			E7		
RVH/Pig-wt/ITA/CR/305471/2017	Lombardia	2	1	1	1	1	P10		G22			I3 clade 2			E3 clade 2		
RVH/Pig-wt/ITA/CR/48625-1/2017	Lombardia	2	2	3	1	3	P12	P10	G13	G18	G22	I3 clade 2			E3 clade 1	E4 clade 1	E4 clade 1
RVH/Pig-wt/ITA/BS/76963/2018	Lombardia	3	2	1	2	1	P12	P10	G13			I7	I3 clade 1		E3 clade 2		
RVH/Pig-wt/ITA/CR/101803/2018	Lombardia	4	1	2	1	1	P10		G13	G22		I3 clade 2			E3 clade 2		
RVH/Pig-wt/ITA/CR/105389/2019	Lombardia	4	1	1	1	1	P12		G20			I3 clade 2			E4 clade 1		
RVH/Pig-wt/ITA/PR/138298/2019	Emilia Romagna	5	1	1	-	1	P12		G13			-			E11		
RVH/Pig-wt/ITA/BS/146160/2018	Lombardia	6	1	2	1	1	P10		G16	G18		I3 clade 1			E4 clade 1		
RVH/Pig-wt/ITA/CR/154488/2018	Lombardia	7	1	1	1	1	P10		G21			I7			E8		
RVH/Pig-wt/ITA/NO/345422/2018	Piemonte	8	1	2	-	-	P12		G14	G16		-			-		
RVH/Pig-wt/ITA/BS/103398/2019	Lombardia	9	1	1	1	1	P12		G20			I3 clade 2			E4 clade 1		
RVH/Pig-wt/ITA/CR/140201/2019	Lombardia	10	1	1	1	1	P7		G22			I3 clade 1			E4 clade 1		
RVH/Pig-wt/ITA/BS/294885/2019	Lombardia	11	-	1	1	1	-		G19			I9			E4 clade 1		
RVH/Pig-wt/ITA/BS/14193-5/2020	Lombardia	12	1	1	1	1	P10		G12			I12			E4 clade 1		
RVH/Pig-wt/ITA/BS/51105-6/2020	Lombardia	12	1	1	1	1	P10/11		G12			I12			E4 clade 1		
RVH/Pig-wt/ITA/BS/25635-4/2020	Lombardia	13	1	3	3	1	P12		G13	G14	G15	I8	I10	I3 clade 1	E3 clade 2		
RVH/Pig-wt/ITA/BS/56594/2020	Lombardia	13	1	2	2	1	P12		G13	G19		I3 clade 1	I3 clade 1		E4 clade 1		
RVH/Pig-wt/ITA/PC/31399/2020	Emilia Romagna	14	-	2	1	1	-		G16	G17		I3 clade 1			E4 clade 1		
RVH/Pig-wt/ITA/CR/43471/2020	Lombardia	15	1	1	1	1	P9		G11			I12			E8		
RVH/Pig-wt/ITA/BG/50439/2020	Lombardia	16	1	1	1	1	P12		G18			I8			E4 clade 1		
RVH/Pig-wt/ITA/CR/66533/2020	Lombardia	17	1	1	1	1	P8		G18			I8			E4 clade 1		
RVH/Pig-wt/ITA/BS/355703/2019	Lombardia	18	1	1	1	1	P12		G18			I10			E10		
RVH/Pig-wt/ITA/BS/42091-1/2020	Lombardia	19	1	2	2	1	P12		G18	G22		I3 clade 1	I3 clade 1		E3 clade 1		
RVH/Pig-wt/ITA/TR/84987/2021	Umbria	20	1	1	1	1	P12		G20			I3 clade 2			ND		
RVH/Pig-wt/ITA/TV/116154/2021	Veneto	21	1	1	1	1	P12		G18			I3 clade 2			E3 clade 2		
RVH/Pig-wt/ITA/CR/165825/2021	Lombardia	22	2	1	-	-	P12	P9	G14			-			-		
RVH/Pig-wt/ITA/BS/243418/2021	Lombardia	23	-	1	2	1	-		G18			I12	I11		E9		

ND, not determined.

**Table 4 viruses-16-00068-t004:** Recombination events detected via the recombination detection program (RDP) v4.101 for RVH VP4.

Strain	Gene	Recombination Breakpoints (bp)	Major Parent	Minor Parent	Detection Method	*p*-Value
ITA/BS/14193-5/2020	VP4	945	ITA/BS/51105-6/2020	ITA154488/2018	RDP	8.48 × 10^3^
				ITA/48625-1(I)/2017	GENECONV	1.88 × 10^−5^
				ITA/101803/2018	Bootscan	1.32 × 10^−4^
				ITA/305471/2017	Maxchi	1.51 × 10^−6^
					Chimaera	3.46 × 10^−9^
					SiSscan	6.18 × 10^−11^
					3 Seq	1.21 × 10^−36^
ITA/BS/51105-6/2020	VP4	945	ITA/BS/14193-5/2020	unknown	RDP	8.48 × 10^3^
					GENECONV	1.88 × 10^−5^
					Bootscan	4.20 × 10^−4^
					Maxchi	3.51 × 10^−2^
					Chimaera	4.18 × 10^−4^
					SiSscan	6.18 × 10^−11^
					3 Seq	1.21 × 10^−36^

## Data Availability

The data presented in this study are contained within the article and Appendix A.

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
