# Peer review of "Identification of Putative Novel Rotavirus H VP7, VP4, VP6 and NSP4 Genotypes in Pigs"

_viruses, 2023, doi:10.3390/v16010068_

Round 1

Reviewer 1 Report

Comments and Suggestions for Authors

This manuscript describes novel RVHs with new genotypes discovered in Italy. The findings obtained in this study include important and interesting points for readers in RV research field. However, the data presented herein is not enough, the revisions need before publication in this journal.

Major revision:

The authors should show genotypes for entire gene of porcine RVHs collected in Italy.

Why did you show their genotypes of limited genes not all genes using NGS?

The authors estimate new genotypes of porcine RVHs in four genes based on re-calculated cut-off values via sequence and phylogenetic analyses. 

The authors should show all data for nucleotide identities in the four genes within and between genotypes including new genotypes in a new Table.

Figure 1

The authors should remake a clear figure.

Author Response

Thank you very much for taking the time to review this manuscript. Please find the detailed responses below and the corresponding corrections highlighted in the re-submitted files.

Comment 1: The authors should show genotypes for entire gene of porcine RVHs collected in Italy.

Why did you show their genotypes of limited genes not all genes using NGS?

We thank the reviewer for this point. We decided to focus on genes encoding proteins VP7, VP4, VP6 and NSP4 because they are crucial for cell-host recognition and pathogenesis.

Moreover, since the present study is retrospective and some samples were partially degraded, we needed to amplify each of these genes by PCR using specific primers. For this reason, the amplification of some gene segments was not possible for all the samples included in the study.

Comment 2: The authors estimate new genotypes of porcine RVHs in four genes based on re-calculated cut-off values via sequence and phylogenetic analyses. 

The authors should show all data for nucleotide identities in the four genes within and between genotypes including new genotypes in a new Table.

We thank the reviewer for the suggestion. We included a supplementary table with intra and inter-genotypes identities of individual genes at the nucleotide level (Supplementary Table S3) and we integrated this information in the text (lines 144-145)

Comment 3: Figure 1

The authors should remake a clear figure.

We are not sure about what might be unclear in the figure 1. The image resolution is compliant with the journal’s requirements. To improve the figure’s clarity, we added additional information in the caption in lines 269-270.

Additional clarifications

In addition to the changes reported to the reviewers we introduced minor changes in the text as follow:

  • we added “genetic” in line 27, “The” in line 33
  • we replaced the sentence “Studies published between 2008 and 2017 have subsequently reported the presence of porcine RVH in United States, Brazil, South Africa, Vietnam and Spain [13–17]” (lines 45-47, original manuscript) with “Between 2008 and 2022 the presence of porcine RVH was attested also in United States [13], Brazil [14], South Africa [15], Vietnam [16], China [17] and some European countries, as Spain [18], Italy [7] and Russia [19]” (lines 45-47, revised manuscript)   
  • We corrected the strain names in lines 182,184,185, 196, 234-235, 237-238, 243-246
  • We changed PX6 in PX5 (line 193)
  • We added I3 clade 1 and clade 2 in lines 220-221
  • We substituted the sentence: ”Two strains (ITA/84987 and ITA/50439) shared percentages of identity above the cut-off with only a few of E4 clade 1 sequences, so their classification is not well defined” (lines 224-225, original manuscript) with “Strain ITA/84987/2021 shared percentages of identity above the cut-off with only a few of E4 clade 1 sequences, so its classification is not well defined” (line 249-250, revised manuscript)
  • We replaced “EX2” in the actual table 3 (line 25 of the table 2) with “ND”.
  • We added the explanation “ND, not determined”, under Table 2 (line 290)
  • We changed the order of words in lines 300 and 302
  • We replaced “For some NSP4 (ITA/84987, ITA/50439) and VP4 (ITA/51105-5) sequences” with “For NSP4 ITA/84987/2021 and VP4 (ITA/51105-6/2020) sequences” in line 332
  • We removed the sentence “This was not unexpected, since confounding factors can contribute to the genetic diversity of viruses, including recombination [32] or convergent evolution [33]. Also, virus evolution via genetic drift may generate a continuum of sequences variants, variously selected and represented in the animal host [34]” and the corresponding references in lines 306-309 of the unrevised version
  • We modified the sentence “We thank Alice Papetti for her support in the phylogenetic analyses with “ We thank Alice Papetti for her support in the phylogenetic analyses and Andrea Boscarino for his support in data curation”, lines 400-401.
  • We removed the previous reference Varghese et al., 2003 and added 2 new references (Krasnikov, N.; Yuzhakov, A., 2023 and Martin, D.P et al., 2015)
  • We modified and added new supplementary materials as follow (lines 378-389)

Table S1: “GenBank accession number of RVH reference strains” was changed with “Supplementary Table S1: Data on porcine farm management and health status” ;

“Table S2: Italian strains used to define nucleotide identity cut-offs” was replaced with “Supplementary Table S2: GenBank accession number of RVH reference (in black) and italian strains (in red) used to calculate nucleotide identity cut-offs”

“Table S3: Nucleotide identity of each gene among RVH strains” was replaced with “Supplementary Table S3: Range of nucleotide identity (%) of each gene among RVH genotypes”

We added: “Supplementary Table S4: Nucleotide identity of each gene among RVH strains.” , Supplementary Table S5: “Recombination events detected by Recombination Detection Program (RDP) v4.101 for RVH VP4” and “Supplementary Figure S1. Similarity analysis of the VP4 gene of the strain ITA/14193-5/2020 and ITA/ 51105-6/2020 with complete RVH VP4 sequences. Strain name of RVH sequences are shown on the right of the plot. The analysis was performed by SimPlot 3.5.1 (10) using a sliding window of 200 nucleotides moving in 20-bp steps”.

Reviewer 2 Report

Comments and Suggestions for Authors

Authors sequenced several genes of RVH, collected over a 4-year period, from 23 pig farms mostly located in Lombardia region, and authors claim that they may have found some novel genotypes of each.

1. the manuscript is too short for an article.

2. the lab work was monotonous (extract nucleic acid from fecal sample, PCR, sequence, desk analysis, most of the works are semiautomatic).  

3. the only significant data is in Table 2.

4. as the authors claimed, the sequence information for RVH is limited, then what is the point of subgenotype to such a detail level?  In other words, what is the purpose of your study? What can you correlate this too lab diagnosis practice (it is not necessary for a veterinary diagnostic lab)?  How can you correlate your results to the diseases found in the field?  Do these novel genotypes that you found related to intestinal diseases other than those caused by more classical genotypes?

5. There is a lack of pig data that authors are studying.  Without this information, the value for reference is limited. Information such as:

- the size of each herd? the farming type? Are they industrial or backyard raising? Are these pigs fed commercialized food or kitchen leftover?  What  is the management of the herd? Is it from farrow-to-finish? or pure fattening pigs.  Or only farrow to weaning. 

- Most important, the age (? weeks old) of the symptomatic pigs collected, are they neonates, weaners,  nursery, or fattening/finishing?  What are the overall vaccination program and the health status.

6. Is it significant, for a virus with segmented genome, to sub-genotyping like what your are doing? It could be endless, unless you establish a "haplotype" like in MHC.

7.  it would be more significant, if authors analyze how and why these gene become so diversified (recombination? recombined with what virus, mutations? etc).  

Author Response

Thank you very much for taking the time to review this manuscript. Please find the detailed responses below and the corresponding corrections highlighted in the re-submitted files.

Comment 1: the manuscript is too short for an article.

We thank the reviewer for pointing this out. The word count conducted on the journal website showed 4100 words, in accordance with the requirement of the journal. With the present revision, we have included additional information in the manuscript (highlighted in yellow), which has extended the text length

Comment 2: the lab work was monotonous (extract nucleic acid from fecal sample, PCR, sequence, desk analysis, most of the works are semiautomatic).  

We are sorry if the lab work sounds monotonous but we employed the necessary methods and instruments to achieve our purpose, which was to sequence porcine rotavirus H strains.

Comment 3: the only significant data is in Table 2.

In fact, Table 2, in conjunction with the phylogenetic trees, serves as a summary of our results, where all the key findings of our study are consolidated.

Comment 4: as the authors claimed, the sequence information for RVH is limited, then what is the point of

We thank the reviewer for the comment and to give us the opportunity to clarify this point. The purpose of the study is to characterize the evolution and variability of porcine RVH strains in Italy. We have significantly enhanced the clarity of the text regarding this aspect in the Introduction section (lines 74-78).

Comment 5: What can you correlate this to lab diagnosis practice (it is not necessary for a veterinary diagnostic lab)?

The lab techniques used in this work are not suitable to the routine lab analyses. However, the generated output data, such as gene sequences, are very useful to upgrade and improve diagnostic assays in order to detect all the circulating variants/genotypes effectively.

Comment 6: How can you correlate your results to the diseases found in the field? 

We thank the reviewer for this suggestion and for the following comment 8. We agree that herd data and information on pig health status are important to better place in context our study. Alle the requested data have been collected in Supplementary Table S1 and detailed in the “Material and Methods” section (lines 92-96). Unfortunately, precise information regarding herd size at the time of the sampling was not available: the farrow to weaning herd ranged between from 500 to 3000 animals, while the weaning herd size ranged between from 2500 to 12000.

Regarding the evaluation of a possible correlation between RVH genotypes and disease, it was not possible to establish it for two reasons: 1) the totality of samples tetsted positive to other rotavirus groups, as reported in Supplementary table S1, making it unclear how each agent contributes to pathogenesis 2) moreover, the sample size is insufficient to infer correlation between genotype and disease. To address this, we've included a discussion on these points in the text (lines 362-366).

Comment 7: Do these novel genotypes that you found related to intestinal diseases other than those caused by more classical genotypes?

As mentioned above, it was not possible to establish correlation between novel genotypes and intestinal disease. Moreover, to date, no functional study has analysed the impact of a specific RVH genotype on intestinal disease.

Comment 8: There is a lack of pig data that authors are studying.  Without this information, the value for reference is limited. Information such as:

- the size of each herd? the farming type? Are they industrial or backyard raising? Are these pigs fed commercialized food or kitchen leftover?  What is the management of the herd? Is it from farrow-to-finish? or pure fattening pigs.  Or only farrow to weaning. 

- Most important, the age (? weeks old) of the symptomatic pigs collected, are they neonates, weaners,  nursery, or fattening/finishing?  What are the overall vaccination program and the health status.

We thank the reviewer for these suggestions. As mentioned above (comment 6), we have, accordingly, supplemented the manuscript with such information in lines 92-96 and in the Supplementary table S1.

Comment 9: Is it significant, for a virus with segmented genome, to sub-genotyping like what your are doing? It could be endless, unless you establish a "haplotype" like in MHC.

We thank the reviewer for the interesting account. At this regard, in our approach we strictly applied the guidelines established and shared by RCWG for the classification of Rotavirus A. This system provides to characterize each single gene, which is the most appropriate method to genotype segmented genomes prone to frequent reassortment events.

Comment 10: it would be more significant, if authors analyze how and why these gene become so diversified (recombination? recombined with what virus, mutations? etc).  

We thank the reviewer for the comment and to give us the opportunity to investigate deeply the diversification of RVH strains. To answer to the reviewer, we added additional data to this study. In particular, we conducted recombination analyses among human and porcine RVH, RVA, RVB, RVC strains. Two recombinant strains were observed in VP4 of RVH (ITA/14193-5/2020 and ITA/51105-6/2020). This data explains the ambiguity, observed in the strain ITA/51105-6/2020, between genotype and phylogenetic tree that unabled a clear genotype classification. No recombination event between RVH and other RV group was detected using the software RDP4. Data on recombination were added in the main text in lines 157-162 (section “Material and Methods”), 197-207 (section “Results”), in Supplementary Table S5, in Supplementary Figure S1, in lines 335-342 (section Discussion).

Additional clarifications

In addition to the changes reported to the reviewers we introduced minor changes in the text as follow:

  • we added “genetic” in line 27, “The” in line 33
  • we replaced the sentence “Studies published between 2008 and 2017 have subsequently reported the presence of porcine RVH in United States, Brazil, South Africa, Vietnam and Spain [13–17]” (lines 45-47, original manuscript) with “Between 2008 and 2022 the presence of porcine RVH was attested also in United States [13], Brazil [14], South Africa [15], Vietnam [16], China [17] and some European countries, as Spain [18], Italy [7] and Russia [19]” (lines 45-47, revised manuscript)   
  • We corrected the strain names in lines 182,184,185, 196, 234-235, 237-238, 243-246
  • We changed PX6 in PX5 (line 193)
  • We added I3 clade 1 and clade 2 in lines 220-221
  • We substituted the sentence: ”Two strains (ITA/84987 and ITA/50439) shared percentages of identity above the cut-off with only a few of E4 clade 1 sequences, so their classification is not well defined” (lines 224-225, original manuscript) with “Strain ITA/84987/2021 shared percentages of identity above the cut-off with only a few of E4 clade 1 sequences, so its classification is not well defined” (line 249-250, revised manuscript)
  • We replaced “EX2” in the actual table 3 (line 25 of the table 2) with “ND”.
  • We added the explanation “ND, not determined”, under Table 2 (line 290)
  • We changed the order of words in lines 300 and 302
  • We replaced “For some NSP4 (ITA/84987, ITA/50439) and VP4 (ITA/51105-5) sequences” with “For NSP4 ITA/84987/2021 and VP4 (ITA/51105-6/2020) sequences” in line 332
  • We removed the sentence “This was not unexpected, since confounding factors can contribute to the genetic diversity of viruses, including recombination [32] or convergent evolution [33]. Also, virus evolution via genetic drift may generate a continuum of sequences variants, variously selected and represented in the animal host [34]” and the corresponding references in lines 306-309 of the unrevised version
  • We modified the sentence “We thank Alice Papetti for her support in the phylogenetic analyses with “ We thank Alice Papetti for her support in the phylogenetic analyses and Andrea Boscarino for his support in data curation”, lines 400-401.
  • We removed the previous reference Varghese et al., 2003 and added 2 new references (Krasnikov, N.; Yuzhakov, A., 2023 and Martin, D.P et al., 2015)
  • We modified and added new supplementary materials as follow (lines 378-389)

Table S1: “GenBank accession number of RVH reference strains” was changed with “Supplementary Table S1: Data on porcine farm management and health status” ;

“Table S2: Italian strains used to define nucleotide identity cut-offs” was replaced with “Supplementary Table S2: GenBank accession number of RVH reference (in black) and italian strains (in red) used to calculate nucleotide identity cut-offs”

“Table S3: Nucleotide identity of each gene among RVH strains” was replaced with “Supplementary Table S3: Range of nucleotide identity (%) of each gene among RVH genotypes”

We added: “Supplementary Table S4: Nucleotide identity of each gene among RVH strains.” , Supplementary Table S5: “Recombination events detected by Recombination Detection Program (RDP) v4.101 for RVH VP4” and “Supplementary Figure S1. Similarity analysis of the VP4 gene of the strain ITA/14193-5/2020 and ITA/ 51105-6/2020 with complete RVH VP4 sequences. Strain name of RVH sequences are shown on the right of the plot. The analysis was performed by SimPlot 3.5.1 (10) using a sliding window of 200 nucleotides moving in 20-bp steps”.

Round 2

Reviewer 1 Report

Comments and Suggestions for Authors

The supplementary Tables showed that the RVH genotypes the authors estimated are classified into new genotypes based on nucleotide identities inter- and intra-genotypes. Therefore, the authors should use a serial number following previous genotypes not X1. Please change the genotypes in the figure and text.

Author Response

Reviewer 1 comment

The supplementary Tables showed that the RVH genotypes the authors estimated are classified into new genotypes based on nucleotide identities inter- and intra-genotypes. Therefore, the authors should use a serial number following previous genotypes not X1. Please change the genotypes in the figure and text.

We agreed with the reviewer observation. We changed genotypes Xn with serial number following the genotypes previously described in literature (Suzuki et al., 2018). According to this suggestion, genotypes from GX1 to GX12 were replaced by G11-G22; genotypes from IX1-IX6 were replaced by I7-I12; genotypes from PX1 to PX6 were replaced by P7-P12 and genotypes from EX1-EX5 were replaced by E7-E11. Therefore, we modified the text (lines 178-179, 181, 183-184, 186, 196, 198, 207, 228, 245, 344 of the revised version), the Figure 1 and Table 4 of the revised version).

Reviewer 2 Report

Comments and Suggestions for Authors

To  improve the value of this manuscript for reference in the veterinary field.  I encourage authors incorporate data supplementary Figure S1, and supplementary Table S1 and Table S5 into the manuscript before publication.

Author Response

Reviewer 2 comment

To  improve the value of this manuscript for reference in the veterinary field.  I encourage authors incorporate data supplementary Figure S1, and supplementary Table S1 and Table S5 into the manuscript before publication.

We thank the reviewer for the suggestions. We provided to integrate Table S1 (Table 1 in this revised version) in the manuscript. Since the table resulted too large, we removed columns “Feeding”, “Vaccination anti-RVA” and “Health status” , which data were equal for all the samples. However, such information is reported in the text in the section “2.1 Samples” (lines 86 and 94-95). In the same section (lines 95-96) we added the sentence “Age data and the status of infection by RV groups, tested by RT-qPCR [7], are reported in the Table 1”.

Regarding the recombination analyses (in Table S5 and Figure S1, in the previous version), we integrated in the manuscript only Table S5 (Table 3, in the revised version) lines 215-217. The Table 3 was modified adding the columns “gene” and “recombination breakpoints” that show the nucleotidic position of the breakpoint.

Since the simplot analysis (fig. S1) is a confirmation  of RDP4 results, we considered more appropriate to keep this data as supplementary material (Figure S1).